# Prevalence of *Schistosoma mansoni* infection among fishermen in Busega district, Tanzania

Revocatus J. L. Mang'ara[1], Billy Ngasala[2], Winfrida John[2,3]*

1 Department of Environmental Health Sciences, Ruaha Catholic University, Iringa, Tanzania, 2 Department of Parasitology and Medical Entomology, School of Public Health and Social Sciences, Muhimbili University of Health and Allied Sciences, Dar es Salaam, Tanzania, 3 National Institute for Medical Research, Dar es Salaam, Tanzania

* jwinrence@gmail.com

**Data Availability Statement:** All relevant data are within the article.

**Funding:** The authors received no specific funding for this work.

## Abstract

### Background

*Schistosoma* (*S.*) *mansoni* infection is endemic in all regions around Lake Victoria and affects all age groups to different degrees. In most endemic areas, less attention has been paid to determining the prevalence of infection, sanitation status, and knowledge about intestinal schistosomiasis (KIS) in fishermen. Therefore, the purpose of this study was to establish the prevalence of *S. mansoni* infection and associated factors among fishermen in the Busega district.

### Materials and methods

A cross-sectional study was conducted among fishermen in July, 2020 in five fishing villages in the Busega district located along Lake Victoria. A total of 352 fishermen were interviewed with regard to their sanitation status and level of KIS. A single stool sample from fishermen was examined for *S.mansoni* eggs by using the Formalin-Ether Concentration technique. The potential factors associated with *S. mansoni* infection were explored using multivariable logistic regression.

### Results

The prevalence of *S. mansoni* infection was high (65.0%) among fishermen and varied with age, whereby fishermen aged ≤36 years had the highest prevalence. Fishermen had a low level of KIS and the majority of them reported practicing open defecation during fishing (81%). These fishermen with a low level of KIS and who reported defecating in open areas during fishing had 2.8 times (95% CI: 1.0–7.2) and 2.1 times (95% CI: 1.1–3.9) higher odds of being infected with *S. mansoni* than those with a high level of KIS and those who did not report defecating in open areas during fishing, respectively.

### Conclusion

*S. mansoni* infection was high among fishermen in the Busega district. Furthermore, fishermen had a low level of KIS and were reported to have defecated in open areas during fishing. Infection with *S. mansoni* was associated with age, a low level of KIS and open

**Competing interests:** The authors have declared that no competing interests exist.

**Abbreviations:** BMU, Beach Management Unit; EPG, Eggs Per Gram; FEC, Formalin-Ether Concentration; GM, Geometric Mean; KIS, Knowledge about intestinal schistosomiasis; KK, Kato-Katz; MDA, Mass Drug Administration; MUHAS, Muhimbili University of Health and Allied Sciences; NBS, National Bureau Of Statistics; NIMR, National Institute for Medical Research; SPSS, Statistical Package for Social Sciences; WHO, World Health Organization.

defecation behaviour during fishing. Therefore, mass drug administration (MDA) with praziquantel, health education, and sanitation behaviour change interventions were needed.

## Background

Schistosomiasis is an acute and chronic parasitic disease caused by blood flukes of the genus *Schistosoma* [1]. Despite efforts to control transmission, more than 220 million people worldwide are infected, with 85% living in Sub-Saharan Africa, where prevalence rates exceed 50% of the local population [1].

Fishermen, irrigation workers, and women carrying out domestic activities are involved in daily contact with infested freshwater and, therefore, are at greater risk of being infected with Schistosoma larvae released by infected freshwater snails [1].

Mild infection can cause abdominal pain, diarrhea, gastrointestinal bleeding, and bloody stools. In advanced cases, it can cause hepatomegaly, splenomegaly, ascites, haematemesis, and varices, which can rapidly lead to death [2]. In fact, schistosomiasis disables more than killing. Disabling complications in children include anaemia, growth stunting, cognitive impairment, and decreased physical fitness [3].

With an estimated prevalence of schistosomiasis of 51.5%, Tanzania is the second country in sub-Saharan Africa to have a high burden of the disease after Nigeria [4]. Poor sanitation, lack of knowledge about schistosomiasis, and activities that involve high water contact like fishing are known to increase the risk of schistosomiasis [4,5].

*S. mansoni* infection is prevalent in all regions around Lake Victoria and affects all age groups to different degrees. The effort to control schistosomiasis is mainly relying on school-based MDA, which excludes adults at risk like fishermen [6,7]. Currently, the magnitude of infection, sanitation status, and KIS on fishermen remains unclear in most endemic areas [8,9]. Therefore, this study was carried out to determine the magnitude of *S.mansoni* infection and associated factors among fishermen living in the Busega district.

## Materials and methods

### Study design

A cross-sectional study was carried out to determine the current magnitude of *S. mansoni* infection and associated factors among fishermen in July, 2020.

### Study population

The study population was fishermen aged above 18 years (an age group not often targeted by MDA) without a history of taking praziquantel within the last 3 months (the incubation period of schistosomiasis), and who provided written informed consent.

### Study area

The study was carried out in Ihale, Nyakaboja, Kalago, Nchilu, and Fogofogo villages in the Busega district. The district was conveniently selected because it had a high prevalence of 79.2% among school-aged children [10]. The district is in the Simiyu region, located on the shores of Lake Victoria between latitude 2° 10' and 2° 50' South and between longitude 33° and 34° East.

Fishing is a major source of income in the district, and there are ten registered fishing boat landing sites in distinct villages such as Ijitu, Ihale, Milambi, Bulima, Nyakaboja, Mayega, Kalemela, Kalago, Nchilu, and Fogofogo [11].

## Sample size

The sample size formula for estimation of a single proportion:

$$n = \frac{Z^2 P(100 - P)}{\varepsilon^2}$$

Where, $n$ = Minimum sample size, $Z$ = Standard normal deviate for given confidence level (CI) = 1.96 for a 95% CI, $p$ = Expected proportion = 29.22% [8] and $\varepsilon$ = Margin of error (the precision) = 5%. Calculated sample size was 318 fishermen. Assuming a 10% non-response, the resulted sample size was 353 fishermen.

## Sampling

Two stage-sampling procedures were used to select villages and individual fishermen. In the first stage, five villages were selected by lottery from a list of ten villages. In the second stage, eligible fishermen from each of five selected villages were selected by a simple random sampling technique using the list of registered names of fishermen as a sampling frame.

The number of sample fishermen from each village was determined by a probability proportional to size formula: n = P*N, where, P = proportion, N = total sample size. The numbers of sample fishermen from each village were as follows: Ihale = (144/1038)353 = 49, Kalago = (106/1038)353 = 36, Nyakaboja = (92/1038)353 = 31, Nyamikoma 'A' = (543/1038)353 = 184, Nyamikoma 'B' = (153/1038)353 = 52.

## Data collection tools and methods

A structured interview schedule was used, adapted from a study conducted in Yemen [12]. The schedule was translated into Kiswahili to make it suitable for the targeted group. The researcher and two trained research assistants collected data, who were accompanied by the Beach Management Unit (BMU) chairperson of the respective landing site. An interview was performed to obtain information on socio-demographic characteristics of fishermen, sanitation behavior during fishing, and their KIS. Furthermore, fishermen were asked about the ownership of latrines, and a direct observation method was used to confirm latrine availability and type.

After the interview, all participants were required to provide only one stool sample in a well-labeled sterile container. The collected samples were mixed with 10% formalin (1 part stool to 3 parts preservative) to preserve the samples and kill fecal pathogens for safety. All collected samples were placed in a box and transported to the NIMR Laboratory at Mwanza Centre, where they were processed and examined the following day.

## Stool examination

An investigator examined collected stools with the assistance of two qualified and experienced laboratory technicians. For each respondent, a single stool specimen was processed using the FEC technique, and double slides were prepared for each sample to be examined under the microscope for *S. mansoni* eggs.

## Data quality assurance

In the course of data gathering, the researcher was making a thorough spot-check of the research assistants to ensure correctness and completeness of the interview schedule and adherence to standard protocols of the checklist. Standard operating procedures (SOP) [13] were followed during specimen collection, transportation, processing, examination, and result recording under the supervision of the researcher. To verify the consistency of microscopic results, 10% of the specimens were randomly selected each day and re-examined by a researcher without prior knowledge of the results.

## Data management and analysis

At the end of each day of data collection, the structured interview schedules were checked for completeness and correctness. The data was then coded before being entered into the Statistical Package for the Social Sciences (SPSS for Windows, version 22.0) and cleaned for errors caused by inconsistent entry. A copy of the data sheet was stored on a separate drive to save as a backup. Then, the recorded data collection sheets were filled in and stored.

KIS was evaluated by five questions with a total of five [5] points. Knowledge scores were categorized as low (<2 points), average (2–3 points) and high (>3 points) [9]. Data on socio-demographic, infection status, latrine ownership, sanitation behaviour, and KIS were summarized as frequencies and proportions then presented in tables and charts.

Pearson's chi-square statistical tests were used to compare proportions between groups. The unadjusted odds ratio (UOR) was estimated by bivariate logistic regression analysis to identify factors associated with *S. mansoni* infection to be included in the multivariate logistic regression. The biological plausible factors and all factors with a P-value < 0.2 in bivariate analysis were included in the final model. Following that, the adjusted odds ratio (AOR) was calculated using enter method of multivariable logistic regression analysis to determine factors that were independently associated with *S. mansoni* infection. The associations were considered significant at P<0.05.

## Ethical considerations

Ethical clearance to carry out the study was obtained before conducting the study from the Institutional Review Board (IRB) of MUHAS (ethics clearance no. IRB#: MUHAS-REC-04-2020-242). Permission to conduct the study in the villages was obtained from the Local authorities of Busega district. Villagers were informed about the study through a village meeting, and written consent was obtained from eligible fishermen prior to recruitment into the study by using a consent form written in Kiswahili.

## Results

### Socio-demographic characteristics of the study participants

Of the 353 eligible participants for the study, 352 fishermen from Ihale, Kalago, Nyakaboja, Nyamikoma 'A', and Nyamikoma 'B' village participated in the study while the one refused to participate. The median age was 35 years with the range of 18 to 74 years. Majority of the participants were males (92.6%), aged ≤36 years (57.4%), married (59.4%), with primary education (84.4%) and resided in Nyamikoma 'A' village (Table 1).

**Table 1. Socio-demographic information of fishermen participated in the study.**

| CHARACTERISTIC | CATEGORY | N = 352 | % |
|---|---|---|---|
| **Age (years)** | ≤36 | 202 | 57.4 |
| | 37–47 | 88 | 25 |
| | 48–58 | 46 | 13.1 |
| | ≥59 | 16 | 4.5 |
| **Gender** | Male | 326 | 92.6 |
| | Female | 26 | 7.4 |
| **Marital status** | Not married | 63 | 17.9 |
| | Married | 209 | 59.4 |
| | Separated/divorced/Widowed | 51 | 14.5 |
| | Cohabiting | 29 | 8.2 |
| **Education level** | Completed/Not completed primary | 297 | 84.4 |
| | Secondary/University | 55 | 15.6 |
| **Village** | Ihale | 49 | 13.9 |
| | Kalago | 36 | 10.2 |
| | Nyakaboja | 31 | 8.8 |
| | Nyamikoma 'A' | 184 | 52.3 |
| | Nyamikoma 'B' | 52 | 14.8 |

## Prevalence of *S. mansoni* infection stratified by socio-demographic characteristics

Majority of respondents agreed to submit their stool samples 309 (87.5%) for examination. The prevalence of *S. mansoni* infection was high 201(65.04%) and varied significantly by age group (P <0.001), ≤36 years being the most affected age group (Table 2).

**Table 2. Prevalence of *S. mansoni* infection in relation to demographic characteristics.**

| VARIABLE | CATEGORY | EXAMINED (N = 309) | POSITIVE (%) | P-VALUE* |
|---|---|---|---|---|
| **Age (years)** | ≤36 | 177 | 133(75.1) | 0.000 |
| | 37–47 | 77 | 41(53.2) | |
| | 48–58 | 40 | 18(45.0) | |
| | ≥59 | 15 | 9(60.0) | |
| **Gender** | Male | 286 | 187(65.4) | 0.662 |
| | Female | 23 | 14(60.9) | |
| **Marital status** | Not married | 54 | 42(77.8) | 0.184 |
| | Married | 183 | 113(61.7) | |
| | Separated/divorced/Widowed | 43 | 28(65.1) | |
| | Cohabiting | 29 | 18(62.1) | |
| **Education level** | Complete/Incomplete primary | 259 | 167 (64.5) | 0.633 |
| | Secondary/University | 50 | 34(68.0) | |
| **Village** | Ihale | 42 | 27 (64.3) | 0.768 |
| | Kalago | 28 | 20 (71.4) | |
| | Nyakaboja | 28 | 16(57.1) | |
| | Nyamikoma 'A' | 165 | 110(66.7) | |
| | Nyamikoma 'B' | 46 | 28 (60.9) | |

*P-value of Pearson Chi-square ($\chi^2$) test.

**Table 3. Knowledge on cause, transmission mode and clinical sign of schistosomiasis.**

| VARIABLE | CATEGORY | N = 352 | % |
|---|---|---|---|
| **Cause of IS** | Worms | 178 | **50.6** |
| | Mosquito | 14 | 3.9 |
| | Snails | 59 | 16.8 |
| | Do not know | 101 | 28.7 |
| **Mode of transmission for IS** | Drinking untreated water | 201 | **57.1** |
| | Eating contaminated food | 8 | 2.3 |
| | Walking barefooted | 7 | 2.0 |
| | Swimming/bathing/fishing in Lake | 82 | 23.3 |
| | Do not know | 54 | 15.3 |
| **The main sign of IS** | Blood in urine | 93 | **26.4** |
| | Blood in stools | 33 | 9.4 |
| | Painful urination | 85 | 24.1 |
| | Stomach ache | 81 | 23.0 |
| | Do not know | 60 | 17.0 |
| **Preventive measure for IS** | Avoid fishing barefooted, swimming or bathing in the lake | 75 | 21.3 |
| | Avoid walking across barefooted | 26 | 7.4 |
| | Wash hands with water and soap | 7 | 2.0 |
| | Avoid drinking untreated water | 147 | **41.8** |
| | Wash fruits before eating | 10 | 2.8 |
| | Do not know | 87 | 24.7 |
| **Treatment of IS** | Traditional medicine | 18 | 5.1 |
| | Hospital medicine | 263 | **74.7** |
| | Do not know | 71 | 20.2 |

**IS** = Intestinal schistosomiasis.

### Level of KIS among fishermen

The level of KIS among fishermen was low (mean knowledge score of 1.97 points). The majority of the fishermen did not know or know incorrect cause (49.4%), mode of transmission (74.7%), manifestation (90.6%) and preventive measure (71.3%) of intestinal schistosomiasis (Table 3).

### Ownership of sanitation facilities in household and sanitation practice during fishing

Majority of fishermen had the locally designed pour flush latrines at their household and few of respondents still are using pit latrines (15.9%). Furthermore, most of the fishermen (81%) declared to defecate in open places during fishing either direct from their fishing boats or in the plastic bag then disposes it in water or in bushes along the lake (Fig 1).

### Factors associated with *S. mansoni* infections among fishermen

In bivariate regression analysis, factors such as age, marital status, level of KIS and sanitation behavior were significantly associated with *S. mansoni* infection. In multivariate regression analysis, factors that were remained to associate with *S. mansoni* infection were age, low level of KIS and open defecation habit. Indeed, compare to those aged ≤ 36 years, fishermen with the age of 37–47 years and 48–58 years had 0.4 times (95% CI = 0.2–0.8) and 0.3 times (95%

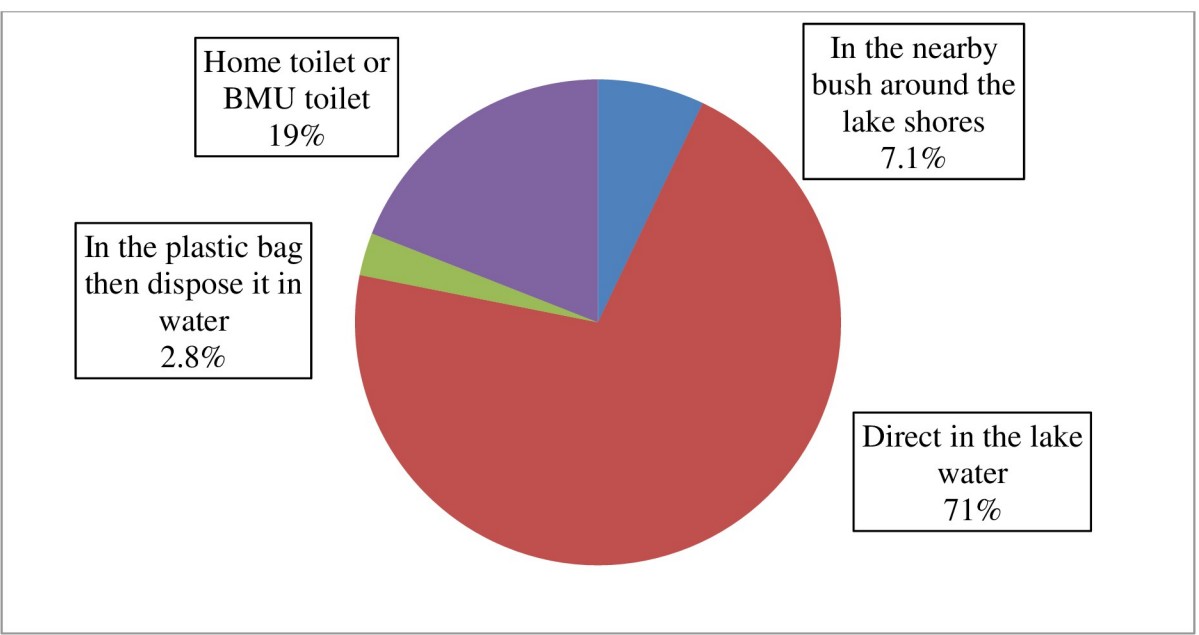

**Fig 1. The place mostly used for defecation during fishing activities.**

CI = 0.2–0.6) less likely to be infected with *S. mansoni* respectively. On the other hand, fishermen with low level of KIS had 2.8 times (95% CI: 1.0–7.2) higher odds of being infected with *S. mansoni* than those with high level of KIS. Equally, fishermen reported to defecate in open areas during fishing had 2.1 times (95% CI: 1.1–3.9) higher odds of being infected with *S. mansoni* than those did not reported to defecate in open areas during fishing (Table 4).

## Discussion

The aim of the present study was to assess the prevalence of *S. mansoni* infection and associated factors among fishermen of the Busega district. We found that fishermen had a high prevalence (65.04%) of *S. mansoni* infection, a low level of KIS, and were reported to practice open defecation during fishing. Furthermore, infection with *S. mansoni* among fishermen was associated with a low level of KIS and open defecation behavior during fishing.

The prevalence observed among fishermen in the Busega district was higher than the prevalence of 51.8% observed among adults living in fishing villages in the Mwanza region [14]. The variations in prevalence might be due to differences in study populations. Furthermore, the prevalence of *S. mansoni* infection among fishermen in the Busega district was higher than those of 29.2% and 15.9% reported among fishermen at Lake Hawassa in Ethiopia [8] and at Alagoasa in Brazil [12] respectively. The difference in prevalence might be due to the variation in climate, ecology, and sanitation practices of the fishermen.

The prevalence we observed in this study varied across the age groups and was higher in those ≤ 36 years old, similar to findings reported from other endemic areas of Tanzania [4]. This may be justified by the change in water contact behaviour among young adults [15]. Young people tend to change recreational swimming or playing in water bodies as they become adults and start carrying out adult roles [16]. This result supports previous studies on the need to include the adult population at risk in de-worming programs since they may serve as a potential reservoir for re-infection of treated schoolchildren [17–19].

**Table 4. Logistic regression analysis of infection status and risk factors.**

| VARIABLE | CATEGORY | UOR(95% CI) | P-VALUE | AOR(95% CI) | P-VALUE |
|---|---|---|---|---|---|
| **Age (years)** | ≤36 | 1 | | 1 | |
| | 37–47 | 0.4 (0.2–0.6) | **0.001** | 0.4 (0.2–0.8) | **0.006**[*] |
| | 48–58 | 0.3 (0.1–0.6) | **<0.001** | 0.3 (0.1–0.6) | **0.001**[*] |
| | ≥59 | 0.5 (0.2–1.5) | 0.207 | 0.5 (0.2–1.5) | 0.218 |
| **Gender** | Male | 1 | | 1 | |
| | Female | 0.8 (0.3–2.0) | 0.663 | 1.3 (0.5–3.5) | 0.577 |
| **Marital status** | Not married | 1 | | 1 | |
| | Married | 0.5 (0.2–0.9) | **0.032** | 0.8 (0.4–1.8) | 0.616 |
| | Separated/divorced/Widowed | 0.5 (0.2–1.3) | 0.17 | 0.9 (0.3–2.3) | 0.775 |
| | Cohabiting | 0.5 (0.2–1.3) | 0.131 | 0.8 (0.3–2.3) | 0.668 |
| **Education level** | Complete/Incomplete primary | 1 | | | |
| | Secondary/University | 1.2 (0.6–2.2) | 0.633 | | |
| **Village** | Ihale | 1 | | | |
| | Kalago | 1.4 (0.5–3.9) | 0.534 | | |
| | Nyakaboja | 0.7 (0.3–2.0) | 0.548 | | |
| | Nyamikoma 'A' | 1.1 (0.55–2.3) | 0.771 | | |
| | Nyamikoma 'B' | 0.9 (0.4–2.1) | 0.741 | | |
| **Knowledge level** | High | 1 | | 1 | |
| | Average | 1.5 (0.6–3.6) | 0.401 | 1.5 (0.6–3.8) | 0.431 |
| | Low | 2.8 (1.1–7.1) | **0.027** | 2.75 (1.0–7.2) | **0.041**[*] |
| **Sanitation behavior during fishing** | Not reported open defecation | 1 | | 1 | |
| | Reported open defecation | 2.2 (1.2–3.9) | **0.009** | 2.1 (1.1–3.9) | **0.026**[*] |

[*]Significant factors with p-value < 0.05, AOR = Adjusted Odds Ratio, UOR = Unadjusted Odds Ratio.

The level of KIS among fishermen was low, and an information gap exists among fishermen because most of them still hold incorrect ideas about the main signs, mode of transmission, and preventive method of intestinal schistosomiasis. The confusion about the signs and symptoms of intestinal schistosomiasis with that of urinary schistosomiasis was similar to a previous study in Swaziland [20]. Furthermore, confusion between mode of transmission and preventive method of schistosomiasis with that of soil-transmitted helminths (STH) seems to be common in many endemic communities of sub-Saharan Africa [5]. However, this misconception about the mode of transmission was found to be less pronounced in some of the schoolchildren [20,21]. As most of the interviewed fishermen were adults, they might not have received health education about schistosomiasis when they were at school.

The majority of fishermen reported defecating in open places during fishing, either in the bushes or directly from their fishing boats. Several studies have found that even a small number of infected people defecating in bodies of water can pose a significant risk of infection to all community members who come into contact with that water later [15,19].

The present study also investigated important risk factors associated with *S. mansoni* infection among fishermen. Age, level of KIS and sanitation habits were factors significantly associated with *S. mansoni* infection among fishermen.

Age was reported to be poor predictor of infection in the previous community study of Kenya [17]. Though in the present study, age was a significant predictor of infection among fishermen, whereby age > 36 years old was associated with decreased infection as compared to

age ≤36 years old, similar to the previous study in Brazil [22,23]. This might be due to leisure-related activities in infected water sources among young fishermen as compared to older ones.

Surprisingly, no significant difference in infection was observed between male and female fishermen. In addition, in this study, gender was not a significant predictor for infection with *S. mansoni*, similar to the previous study in Kenya [17]. However, in a previous study from Tanzania, males were more exposed and infected than females during fishing [4].

Furthermore, the present study showed that the sanitation behaviour of fishermen was a significant predictor of infection, whereby fishermen who reported practicing open defecation either in bushes or in lakes had increased infection compared to those who did not report open defecation practice during fishing. The findings were consistent with previous results in Uganda [24] and Kenya [17]. The majority of fishermen reported defecating openly during fishing and evidence of faecal materials disposed in bushes along the lake were observed during data collection. The practice of open defecation in bushes might be responsible for the observed prevalence of infection among fishermen living in these areas. Therefore, decreasing community-wide open defecation practices can lower the prevalence of *S. mansoni* infection in the fishing community [25].

In this study, fishermen with a low level of KIS were at higher odds of being infected with *S. mansoni* than those with a high level of KIS. Therefore, provision of health education to this community can reduce the prevalence of infection [25]. Our findings are consistent with those reported in Nigeria [26]. However, in Cameroon, people with a high level of KIS were found to be at higher odds of being infected by *S. mansoni* than those without [27]. This might be due to the fact that health education provided to the people did not allow them to change their behaviour to prevent re-infection.

## Limitations

The present study was subject to some limitations. Stool samples collected only on a single day to examine *S. mansoni* may have underestimated the prevalence of infection in the study population as parasite egg output fluctuates day to day. Therefore, future population studies may enhance this by collecting stool samples for at least two consecutive days. Consequently, about 12.5% of fishermen failed to provide stool samples hence the prevalence of infection may have been underestimated.

Furthermore, defecation behaviour during fishing was self-reported, which may have underestimated the actual proportion of individuals practicing open defecation in the study area.

## Conclusion

In conclusion, the prevalence of *S. mansoni* infection among fishermen in the Busega district was high. The prevalence of infection varied with the age of the fishermen whereby young fishermen (aged ≤36 years) had the highest prevalence. Fishermen had a low level of KIS and practiced open defecation during fishing. Infection with *S.mansoni* was associated with a low level of KIS and open defecation behaviour during fishing.

The results of this study support the call for the inclusion of fishermen in the population targeted by the MDA program with praziquantel. This is important since infected fishermen may serve as potential reservoirs of *S. mansoni* infection and might be responsible for re-infection of treated school-aged children as well as transmission of *S. mansoni* to other community groups. Also, health education should be provided in the fishing community as a supplement to MDA programs to address misconceptions about the mode of transmission, symptoms, and prevention of intestinal schistosomiasis.

## Acknowledgments

We would like to express our honest appreciation to all fishermen of Busega who participated in this study, together with chairperson of Ihale, Nyamikoma A and B, Nchilu and Kalago Beach management units as well as Busega district council for allowing us to carry out this study.

The special thanks goes to laboratory technicians from the National Institute for Medical Research (NIMR) Mwanza Centre, Mr. Tupevilwe Mbilinyi, Elias John and Devis Lyatuu for technical support during laboratory examinations of stool samples and their support during data collection.

## Author Contributions

**Conceptualization:** Revocatus J. L. Mang'ara, Billy Ngasala, Winfrida John.

**Data curation:** Revocatus J. L. Mang'ara, Billy Ngasala, Winfrida John.

**Formal analysis:** Revocatus J. L. Mang'ara, Billy Ngasala, Winfrida John.

**Funding acquisition:** Revocatus J. L. Mang'ara.

**Investigation:** Revocatus J. L. Mang'ara, Billy Ngasala.

**Methodology:** Revocatus J. L. Mang'ara, Billy Ngasala, Winfrida John.

**Project administration:** Revocatus J. L. Mang'ara.

**Resources:** Revocatus J. L. Mang'ara.

**Software:** Revocatus J. L. Mang'ara.

**Supervision:** Revocatus J. L. Mang'ara, Billy Ngasala.

**Validation:** Revocatus J. L. Mang'ara.

**Visualization:** Revocatus J. L. Mang'ara, Winfrida John.

**Writing – original draft:** Revocatus J. L. Mang'ara, Billy Ngasala, Winfrida John.

**Writing – review & editing:** Revocatus J. L. Mang'ara, Billy Ngasala, Winfrida John.

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
