## [Decision Letter · Decision Letter 0]

28 Jul 2022

PONE-D-22-14442Prevalence of Schistosoma mansoni infection among fishermen in Busega district, TanzaniaPLOS ONE

Dear Dr. Winfrida John,

Thank you for submitting your manuscript ‘**Prevalence of Schistosoma mansoni infection among fishermen in Busega district, Tanzania**’ to PLOS ONE. Your manuscript has been assessed by 2 reviewers. There are useful comments and suggestions to be improved the manuscript.  One of the reviewers have highligned several major concerns and questions in the sanitized file.   Therefore, we invite you to submit a revised version of the manuscript that addresses the points raised during the review process.

We look forward to receiving your revised manuscript.

Kind regards,

Wannaporn Ittiprasert, Ph.D

Academic Editor

PLOS ONE

Journal Requirements:

Reviewers' comments:

Reviewer's Responses to Questions

**Comments to the Author**

1. Is the manuscript technically sound, and do the data support the conclusions?

Reviewer #1: Yes

Reviewer #2: Yes

2. Has the statistical analysis been performed appropriately and rigorously? 

Reviewer #1: Yes

Reviewer #2: Yes

3. Have the authors made all data underlying the findings in their manuscript fully available?

Reviewer #1: Yes

Reviewer #2: Yes

4. Is the manuscript presented in an intelligible fashion and written in standard English?

Reviewer #1: Yes

Reviewer #2: Yes

5. Review Comments to the Author

Reviewer #1: The paper seeks to address a topical issue of schistosomiasis among the fishermen in Tanzania. The paper determines the prevalence and risk factors for Schistosoma mansoni infection among fishermen in Tanzania. This is timely and very relevant as most studies focus on school-aged children. However, the authors need to address some issues highlighted in the reviewed manuscript before the paper can be considered for publication.

Reviewer #2: Comments to the Author

This manuscript reviews and assesses the current situation of people infected with Schistosoma mansoni in Busega district, Tanzania. Very interesting correlations between socioeconomic status, education, and infection levels were found in this study. Whereby the group of fishermen, a group which is at high risk of getting infected by schistosomes was investigated in more detail. The manuscript elaborated on the differences in sanitation access and management, highlighting the role of fishermen as reservoir hosts. This highlighted that treatment of only the group of pupils with praziquantel was not sufficient to control the current infection events. In order to record infection spreads more precisely, it would have been very interesting to also record the fishing-routes of selected infected and healthy fishermen. Furthermore, an additional figure containing a map showing the exact positions of sampled persons would be desirable to provide a better visual overview.

Especially the significant connection between the age and the status of infection was demonstrated in an impressive way. In addition, it was very delightful to get more information about the correlation of current infections and social status, like marital status, education level etc. However, the resolution to distinguish the different levels of educational level could be more detailed.

Weaknesses of the statistical analysis and sampling were critically discussed in the “Limitations” chapter. The discussed outlook, how the protocol can be adapted in future studies, showed a critical analysis of the own data.

The data presented, clearly underline the conclusions, which were made. Likewise, the conduct of the study, such as survey execution and evaluation, as well as the process of sample preparation were described in a technically clear and detailed manner. Subsequently, the statistics were described in an appropriate manner.

The presented data clearly show that education, social status and hygiene play an essential role in the occurrence of infections. This demonstrated that educational campaigns are essential in the fight against infectious diseases such as schistosomiasis. In addition, the manuscript was written in clear, scientific language.

6. PLOS authors have the option to publish the peer review history of their article (what does this mean?). If published, this will include your full peer review and any attached files.

Reviewer #1: No

Reviewer #2: No

---

## [Author Response · Author response to Decision Letter 0]

28 Aug 2022

Thank you for the comments and suggestions. The following are response to the comments given by the editor and also the reviewers.

-This manuscript is written according to the PLOS one’s style requirement.

-A thoroughly copyedited manuscript is done accordingly.

-The abstract has been amended as suggested.

-The global information on schistosomiasis is included as suggested.

-This age group (18 years and above) is not often targeted by MDA.

-Due to time constraints, data on the frequency of fishermen's contact with water were not collected.

-The stool samples were collected and processed the next day, so they were preserved for no more than 24 hours.

-The enter method was used in the data regression analysis.

-For a variety of reasons, including a lack of time for data collection, approximately 12.5% of fishermen did not provide stool samples. This -concern has been incorporated into the study's limitations.

-There were 201 infected individuals (65.04%). This information has been incorporated across the document.

---

## [Decision Letter · Decision Letter 1]

6 Oct 2022

Prevalence of Schistosoma mansoni infection among fishermen in Busega district, Tanzania

PONE-D-22-14442R1

Dear Dr. Winfrida John,

We’re pleased to inform you that your manuscript has been judged scientifically suitable for publication and will be formally accepted for publication once it meets all outstanding technical requirements.

Kind regards,

Wannaporn Ittiprasert, Ph.D

Academic Editor

PLOS ONE

Additional Editor Comments (optional):

Thank you for revising the manuscript to address the reviewer's concerns. This study contributes to our understanding in the Schistosoma mansoni infection among fishermen in the epidemic area, and should be informative to the field.

Reviewers' comments:

Reviewer's Responses to Questions

**Comments to the Author**

1. If the authors have adequately addressed your comments raised in a previous round of review and you feel that this manuscript is now acceptable for publication, you may indicate that here to bypass the “Comments to the Author” section, enter your conflict of interest statement in the “Confidential to Editor” section, and submit your "Accept" recommendation.

Reviewer #1: All comments have been addressed

Reviewer #2: All comments have been addressed

2. Is the manuscript technically sound, and do the data support the conclusions?

Reviewer #1: Yes

Reviewer #2: Yes

3. Has the statistical analysis been performed appropriately and rigorously? 

Reviewer #1: Yes

Reviewer #2: Yes

4. Have the authors made all data underlying the findings in their manuscript fully available?

Reviewer #1: Yes

Reviewer #2: Yes

5. Is the manuscript presented in an intelligible fashion and written in standard English?

Reviewer #1: Yes

Reviewer #2: Yes

6. Review Comments to the Author

Reviewer #1: (No Response)

Reviewer #2: This manuscript reviews and assesses the current situation of people infected with Schistosoma mansoni in Busega district, Tanzania. Whereby the group of fishermen, a group which is at high risk of getting infected by schistosomes was investigated in more detail.

In general, the points previously noted have been clarified and revised in this manuscript.

In order to record infection spreads more precisely, it would have been very interesting to also record the fishing-routes of selected infected and healthy fishermen. Furthermore, an additional figure containing a map showing the exact positions of sampled persons would be desirable to provide a better visual overview. Visualizing these data, as well as locating the various anchorages of fishers, could improve the data with further clues in terms of the epidemiology. These data would be of great interest to include in a follow-up publication.

With the additions, the Conclusion was made more comprehensible. In addition, the importance and lack of knowledge of the 18+ age group was explored in more detail and depth.

7. PLOS authors have the option to publish the peer review history of their article (what does this mean?). If published, this will include your full peer review and any attached files.

Reviewer #1: No

Reviewer #2: No

---

## [Editor Report · Acceptance letter]

26 Oct 2022

PONE-D-22-14442R1 

Prevalence of *Schistosoma mansoni* infection among fishermen in Busega district, Tanzania 

Dear Dr. John:

I'm pleased to inform you that your manuscript has been deemed suitable for publication in PLOS ONE. Congratulations! Your manuscript is now with our production department. 

Kind regards, 

on behalf of

Dr. Wannaporn Ittiprasert 

Academic Editor

PLOS ONE